ecology/microbiology

*Coptotermes curvignathus*, *Serratia marcescens*, *Pseudomonas aeruginosa*, bacterial transfer medium, application strategies

**Authors for correspondence:**
Kit Ling Chin
e-mail: kitling.chin419@gmail.com
Paik San H'ng
e-mail: ngpaiksan@gmail.com

# Application strategies by selective medium treated with entomopathogenic bacteria *Serratia marcescens* and *Pseudomonas aeruginosa* as potential biocontrol against *Coptotermes curvignathus*

Kit Ling Chin[1], Paik San H'ng[1,2], Chuan Li Lee[1], Wan Zhen Wong[2], Wen Ze Go[2], Pui San Khoo[1], Abdullah Chuah Luqman[1] and Zaidon Ashaari[2]

[1]Institute of Tropical Forestry and Forest Product, and [2]Faculty of Forestry and Environment, Universiti Putra Malaysia, 43400 Serdang, Selangor, Malaysia

CLL, 0000-0002-2293-4088; PSK, 0000-0002-8584-9198

The success of microbial termiticides in controlling termites depends on the ability of microbes to grow in different media and the functionality of the microbes as a resistant barrier or toxic bait. This study was conducted to understand the mortality rate and behaviour changes of the subterranean termite *Coptotermes curvignathus* Holmgren introduced with different concentrations of *Serratia marcescens* strain LGMS 1 and *Pseudomonas aeruginosa* strain LGMS 3 using wood and soil as bacterial transfer medium. In general, higher concentration of bacteria in soil caused a reduction in tunnelling activity and wood consumption and an increase in mortality. However, application on wood revealed a different outcome. Wood treated with *S. marcescens* of $10^6$ CFU ml$^{-1}$ concentration proved to be more efficient as bait than higher concentration applications as it caused a high mortality rate while still highly palatable for termites. Wood or soil treated with *S. marcescens* concentration higher than $10^9$ CFU ml$^{-1}$ creates a high toxicity and repellent barrier for termites. *Pseudomonas aeruginosa* of $10^9$ CFU ml$^{-1}$ concentrations applied on wood served as a

# 1. Introduction

Termite damage is a major problem in the tropics that causes a wide array of damage to plantations, trees and man-made structures. Economic losses in nurseries and during the establishment of various types of agricultural crops and forest plantations can be substantial. In 2018, termites caused an estimated USD 40 billion in damages which accounted for 19% of the total global pest control market [1]. Subterranean termites, particularly species of *Coptotermes*, were reported as the most destructive subterranean termite in Southeast Asia. Species of *Coptotermes* in Malaysia cause more than 85% of total building and structures infestations [2]. Among *Coptotermes* spp. in Malaysia, *Coptotermes curvignathus* Holmgren is the most widespread and aggressive species which attacks buildings and tree plantations, especially rubber, coconut and oil palm [3].

Subterranean termites such as *C. curvignathus* build their nests in the soil underground and prefer different conditions than drywood termites that build their nests in the wood they infest. These unique behavioural differences must be addressed with application strategies to effectively control the infestation of termites. Conventional termiticide treatments are delivered through sprays and fumigation. These treatments either create a repellent barrier to termite activity or serve as contact toxicants. In recent years, baiting systems have been introduced using toxicants that act as chitin synthesis inhibitors. Bait treatments may be able to control a much larger population of subterranean termites. Subterranean termites' colonies could be located hundreds of feet from the infested area and may be 15–20 feet deep in the ground where termiticide spray method may not be able to penetrate deep into the nest to create contact intoxication but rather as a repellent barrier to termite entry. However, the bait method may be able to attract termites to feed on it and cause intoxication through ingestion. Foraging termites will ingest toxins within the bait and then carry these toxins back with them to the colony, where they will share the lethal bait with their nest-mates [4].

The conventional termiticide methods may be replaced with biocontrol, by replacing the conventional toxic chemicals used for termite control with microbial termiticides. Biological control (biocontrol) can be defined as the control or reduction of termite damage by declining the quantity and detrimental activity of termites using the entomopathogenic microorganism or the product resulting from natural biological process. With successful implementation, it can create pest populations management which is sustainable, cost-efficient and with minimal ecological disruption [5]. However, the application of microbial termiticides involves a complex interaction between termite and the microbe.

The success of microbial termiticides in controlling termites depends on the ability of microbes to grow in different media and the functionality of the microbes as a resistant barrier or toxic bait. Microbial termiticides through soil application can only be effective if they cause contact intoxication to termites, while termiticide bait methods will only work if the microbial termiticide creates intoxication through ingestion. Both application strategies of microbial termiticides aim to infect the termite with the pathogen and further spread to other nest-mates through grooming. Through a trophallaxis transfer effect, foraging termites that come into direct contact with microbial termiticides are expected to transfer the pathogen to other termites within the nest through feeding and grooming [6].

Two entomopathogenic bacteria, *Serratia marcescens* and *Pseudomonas aeruginosa*, isolated from the *C. curvignathus*-infested soil in our previous research [7,8], were selected as the potential microbial termiticide in this study. *Pseudomonas aeruginosa* and *S. marcescens* are worth further examination as a possible biocontrol agent against *C. curvignathus*, as both bacteria were reported to be able to disturb termite gut ecology and have the capability to kill termites, even through physical contact with the bacteria [8]. *Serratia* spp. and *Pseudomonas* spp. have been shown to have the ability to produce different degrading enzymes and metabolites that may cause septicaemia on different insects [9–13]. A variety of toxins and extracellular degradative enzymes, such as protease, lipase, carbohydrase and chitinase, have been reported to be secreted by *S. marcescens* that allow this bacterium to completely degrade and exploit the whole body of the insect host [14,15]. The presence of these virulence factors could assist the *S. marcescens* isolate to infect the insects, particularly through feeding infection as the bacterium persists in the gut of insects and crosses the intestinal barrier to reach the haemocoel. Toxic compounds produced by the *P. aeruginosa* such as extracellular proteinases and metalloproteases are

exported throughout the insect host's body as a result of intestinal infection [16–18]. Active enzymes (metalloproteases alkaline proteinase, Prt protease) degrade insect tissues during bioconversion of peptides and amino acids to smaller sulfur, nitrogen and phosphorus compounds [19]. Some selected strains of *S. marcescens* and *P. aeruginosa* have also been found to reduce plant disease severity to a desirable extent using specific application strategies [20–28].

The transmission and bioavailability of pathogen for a period of time depends on the applied medium as an effective growth medium for the microbes. Cellulosic materials which are preferred by a specific type of termites are commonly used as termites' bait. It is essential to identify the availability of pathogens on the application medium to retain a useful level of toxicity on termites over a period of time. However, the termiticide effects and bioavailability of *S. marcescens* and *P. aeruginosa* on different application media over a period of time were not known. Thus, this study was carried out to examine the efficacy of *S. marcescens* and *P. aeruginosa* against subterranean termite *C. curvignathus* in two different application media; bacteria-treated soil and bacteria-treated wood.

This study was conducted to understand the mortality rate and behaviour changes of *C. curvignathus* introduced with different bacterial concentrations using wood and soil as microbial transfer media. Both media were expected to transfer the bacterial infection on to termites in different ways and create toxicity that causes mortality to *C. curvignathus*. Consequently, the aim of the present study was to investigate the efficient method to introduce *S. marcescens* and *P. aeruginosa* as a microbial termiticide for *C. curvignathus* under laboratory conditions.

# 2. Material and methods

## 2.1. Bacteria suspension

Two bacterial strains *S. marcescens* strain LGMS 1 (Accession no.: KP676148) and *Pseudomonas* strain *aeruginosa* strain LGMS 3 (Accession no.: KT725249) used in this study were isolated from *C. curvignathus*-infested soil at a rubber (*Hevea brasiliensis*) plantation of Rubber Research Institute of Malaysia (RRIM), Sungai Buloh (3.16° N, 101.56° E). The detail of isolation, screening and identification was reported in our previous study [7]. Isolated bacterial strains were maintained as stock cultures and were stored on a cool, low-light shelf. These stock cultures were maintained by re-plating each onto new nutrient media at least once a month depending on the nature of each isolated strain.

Bacterial strains from stock cultures were later transferred to fresh Luria bertani (LB) (BD Difco™) agar to obtain a single colony for each isolate and incubated at 28°C. Bacteria were then transferred and cultured in 500 ml Erlemeyer flasks containing 100 ml of LB broth (BD Difco™), incubated in a rotary shaker at 150 r.p.m. and maintained at a temperature of 28°C for 72 h. The suspensions of bacteria were prepared at concentrations of 10, $10^3$, $10^6$, $10^9$ and $10^{12}$ CFU ml$^{-1}$ according to the method published by Brown [29]. At the same time, the purity and colony-forming units for the cultures were checked to confirm their concentration by using pour-plating and spread-plating methods.

## 2.2. Termites

Subterranean termites, *C. curvignathus*, were collected from termite-infested rubber tree (*Hevea brasiliensis*) at the campus of Universiti Putra Malaysia, Selangor. A container with drilled holes containing rubber tree's wooden stakes was buried under the tree to bait and trap foraging termites. After 2 days, the container with infested stakes was removed and transported back to the laboratory. Debris attached on termites was separated using the bridging method as published by Tamashiro *et al.* [30].

## 2.3. Medium preparation

### 2.3.1. Wood block

Untreated rubber wood was collected from Lembaga Getah Malaysia (LGM) and cut into blocks measuring 25 mm square by 5 mm in the tangential direction and autoclaved at 121°C for 1 h. The wood blocks were then oven-dried at 100°C for 3 days, cooled in a desiccator and weighed. The dried sterile wood blocks were later soaked in 50 ml of bacteria suspension (10, $10^3$, $10^6$, $10^9$ and $10^{12}$ CFU ml$^{-1}$) for 24 h. Each block was then removed from the bacteria suspension and placed on a sterile glass dish to let it air-dry for 30 min in laminar air flow.

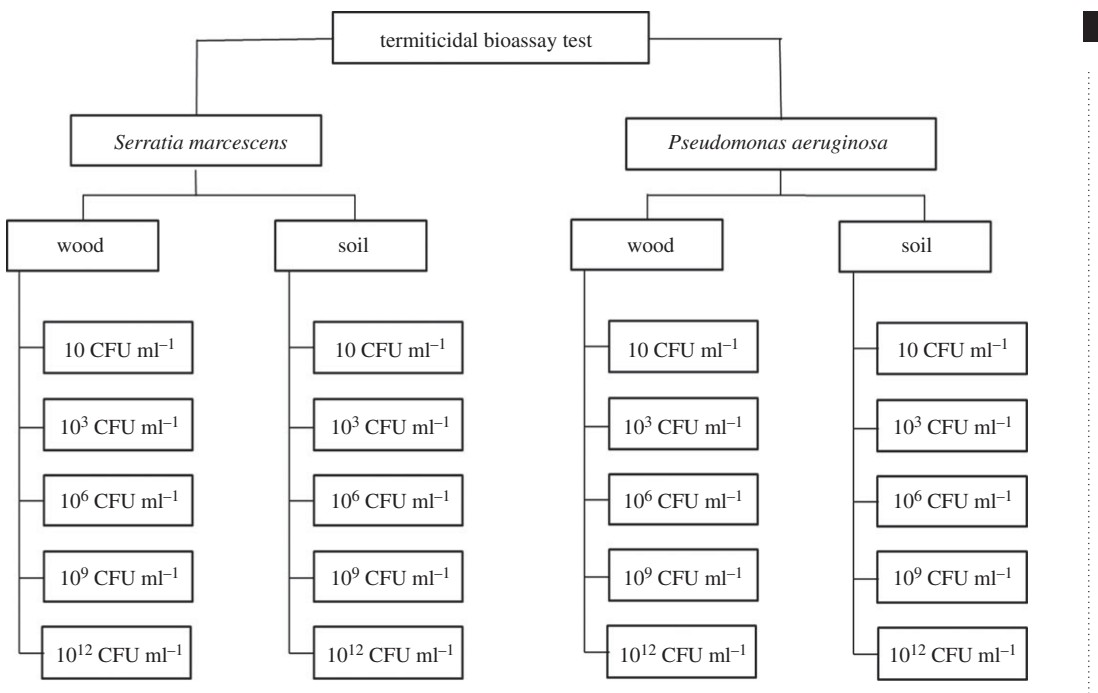

**Figure 1.** Experimental flow for the study to examine the efficacy of two selective media treated with *S. marcescens* and *P. aeruginosa* against *C. curvignathus*.

### 2.3.2. Sand

The washed and oven-dried sand was autoclaved at 121°C for 60 min referring to the method stated by San & Ling [31]. These procedures were applied to remove organic matter and potentially harmful bacteria in the sand that might affect the result of the test. One hundred and fifty grams of autoclaved sand were added with bacteria suspension of different concentration (10, $10^3$, $10^6$, $10^9$ and $10^{12}$ CFU ml$^{-1}$). The amount of bacteria suspension (percentage of dry sand mass) to be added to the sand was calculated using the following formula according to American Wood Preservers' Association standard (AWPA E1-09) [32]:

$$\text{water holding capacity (\%)} = \frac{\text{(weight of water)}}{\text{(oven dry weight of sand)}} \times 100$$

$$\frac{\text{amount of water}}{\text{bacteria suspension to be added (\%)}} = \text{water holding capacity (\%)} - 7\%$$

The sand water holding capacity (saturation point) is defined as the point when the addition of more water results in free water on the surface of sand.

## 2.4. Termiticidal bioassay test assembly

The procedure for bioassay was modified from the American Wood Preservers' Association (AWPA) standard E1-09 [23]. Two selective media (soil and wood) inoculated with five different bacterial concentrations (10, $10^3$, $10^6$, $10^9$ and $10^{12}$ CFU ml$^{-1}$) were used in this study. The experimental design is summarized in figure 1. All treatments were replicated six times.

For wood application strategy, rubberwood block treated with bacteria was placed in a round glass container (80 mm diameter, 75 mm height) containing 150 g autoclaved sand (121°C for 60 min) with 7% moisture content below water holding capacity. For sand application strategy, 150 g of sand containing bacteria suspension of different concentration was placed in the sterilized glass jar. The bacteria suspension functioned as bacteria supply and moisture, while autoclaved rubberwood block was placed in the glass container as the food source in sand application strategy. Control was prepared by placing autoclaved rubberwood in a glass container with 150 g autoclaved sand (7% moisture content under water holding capacity). No bacteria were introduced to any of the media (soil and wood) for

**Table 1.** Rating system for evaluations of termite mortality, behavioural observations and wood block mass loss.

| assessment item | evaluation/rating system |
| --- | --- |
| per cent termite mortality (according to AWPA [32]) | none: 0%<br>slight: 1–33%<br>moderate: 34–66%<br>heavy: 67–99%<br>complete: 100% |
| visual evaluation on the tunnelling activities based on rating scale 0–4 (according to Yeoh & Lee [33]) | 0: no tunnelling activity<br>1: tunnelling activities less than or equal to 25% of total container area)<br>2: 26–50% of total container area<br>3: 51–75% of total container area<br>4: tunnelling activities greater than or equal to 75% of total container area |
| evaluation of termite resistance classes based on wood weight losses (adopted from SNI [34]) | very resistant: less than 3.52%<br>resistant: 3.52–7.50%<br>moderate: 7.50–10.96%<br>poor: 10.96–18.94%<br>very poor: greater than 18.94% |
| visual rating of termite attack on wood blocks based on rating scale 0–10 (according to AWPA [32]) | 10: sound<br>9.5: trace, surface nibbles permitted<br>9: slight attack up to 3% of cross-sectional area affected<br>8: moderate attack, 3–10% of cross-sectional area affected<br>7: moderate/severe attack, penetration, 10–30% of cross-sectional area affected)<br>6: severe attack, 30–50% of cross-sectional area affected<br>4: very severe attack, 50–70% of cross-sectional area affected<br>0: failure |

control samples. Two hundred and fifty workers and 25 soldiers of *C. curvignathus* were later released in each of the assembled glass containers and kept in an incubator at 28°C in total darkness for 28 days. These glass containers were kept in a moist condition (70–80% relative humidity) for four weeks.

## 2.5. Evaluation

These containers were examined every 7 days for 28 days, to record the presence of tunnelling and position of the termites in the container. At the end of the experiment, all wood blocks were removed and oven-dried at 90°C for 3 days and allowed to cool to room temperature in a desiccator for 1 h before weighing. The mortality of the termites was also recorded at the end of the experiment. Termite mortality, behavioural observations and wood block mass loss were recorded using the rating system, as shown in table 1.

## 2.6. Data analysis

All statistical analyses were performed using the SAS software (v. 9.0). The data on termite mortality, wood block visual rating and weight loss for each treatment were analysed using ANOVA at a 95% confidence level ($p < 0.05$). The Tukey–Kramer's multiple comparison test was applied to analyse the differences of the bacterial suspension concentration effect when significance was observed. The effects were considered not statistically significant when the *p*-value was higher than 0.05 at a 95% confidence level.

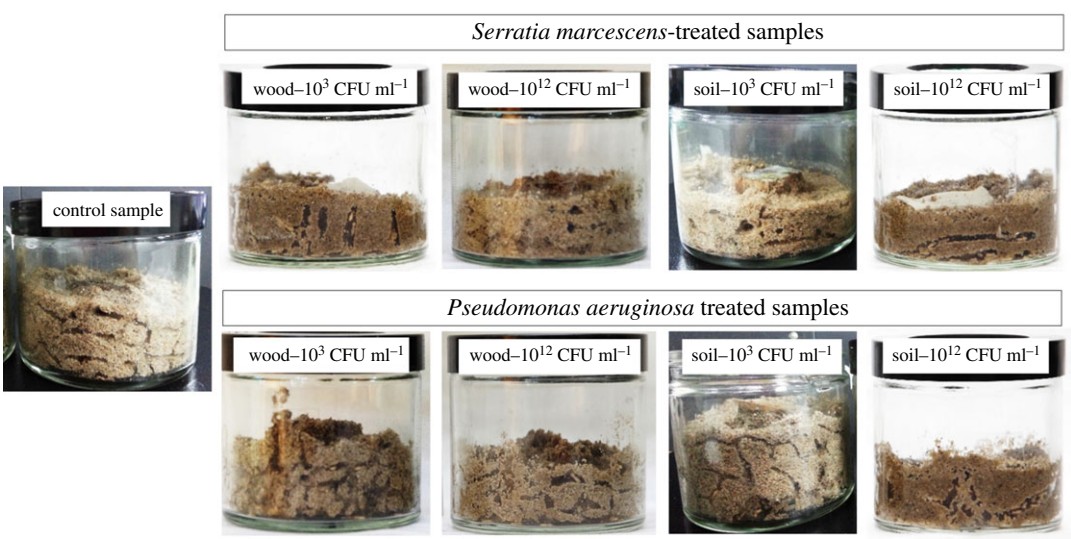

**Figure 2.** Tunnelling activity of *C. curvignathus* in the control sample and different media treated with *S. marcescens* and *P. aeruginosa* in the fourth week (Day 28).

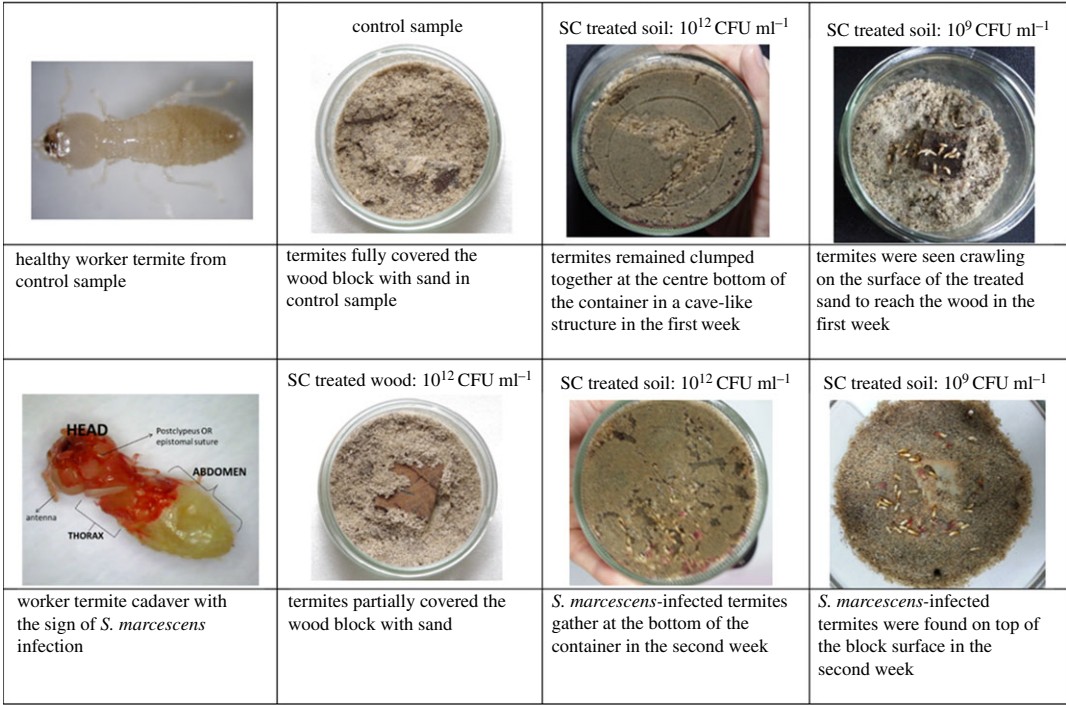

**Figure 3.** Behavioural activities of *C. curvignathus* in control samples and different media treated with *S. marcescens* (SC).

# 3. Results and discussion

The results demonstrated the effects of bacterial suspension concentration and application methods on the behaviour and mortality of *C. curvignathus*. Tunnelling activity and behavioural activities of termites in the control sample and different media treated with *S. marcescens* and *P. aeruginosa* are shown in figures 2 and 3. The mortality of the termite in response to the application of *S. marcescens* and *P. aeruginosa* with different concentrations on different transfer media were reported.

## 3.1. Response of *C. curvignathus* to selective media cultured with *S. marcescens*

### 3.1.1. *Serratia marcescens* cultured on wood

The ANOVA analyses for wood weight loss, visual rating and mortality rate of *C. curvignathus* on wood block treated with different bacterial concentrations of *S. marcescens* are summarized in table 2. Termite

**Table 2.** Behavioural activities and mortality rate of *C. curvignathus* on wood block treated with different bacterial concentrations of *S. marcescens.* Values in parentheses are standard deviations. [a–f]Means followed by the same letter in the same column are not significantly different $p \leq 0.05$ according to Tukey's multiple comparison test.

| concentration (CFU ml$^{-1}$) | week | visual rating of tunnelling behaviour | per cent termite mortality [rating] | percentage (%) of wood weight loss [resistance classes] | mean visual rating of wood block | traceability of *S. marcescens* |
|---|---|---|---|---|---|---|
| control | 1st | 3 | 16.9 (1.1)[d] | 37.0 (1.2)[f] | 2.3 (0.8)[d] | no |
| | 2nd | 4 | [slight] | [very poor] | | no |
| | 3rd | 4 | | | | no |
| | 4th | 4 | | | | no |
| 10$^{12}$ | 1st | 3 | 76.4 (6.7)[a] | 7.1 (1.7)[a] | 9.3 (0.8)[a] | yes |
| | 2nd | 3 | [heavy] | [resistant] | | yes |
| | 3rd | 3 | | | | yes |
| | 4th | 3 | | | | no |
| 10$^9$ | 1st | 3 | 75.1 (4.9)[a] | 9.4 (1.3)[b] | 8.8 (0.5)[a] | yes |
| | 2nd | 3 | [heavy] | [moderate] | | yes |
| | 3rd | 3 | | | | no |
| | 4th | 3 | | | | no |
| 10$^6$ | 1st | 3 | 79.2 (4.1)[a] | 15.1 (0.7)[c] | 6.8 (0.4)[b] | yes |
| | 2nd | 3 | [heavy] | [poor] | | yes |
| | 3rd | 3 | | | | no |
| | 4th | 3 | | | | no |
| 10$^3$ | 1st | 3 | 61.2 (1.5)[b] | 17.8 (0.6)[d] | 6.3 (0.4)[b] | yes |
| | 2nd | 3 | [moderate] | [poor] | | no |
| | 3rd | 4 | | | | no |
| | 4th | 4 | | | | no |
| 10 | 1st | 3 | 50.7 (1.5)[c] | 22.2 (0.7)[e] | 4.3 (0.4)[c] | yes |
| | 2nd | 4 | [moderate] | [very poor] | | no |
| | 3rd | 4 | | | | no |
| | 4th | 4 | | | | no |
| *p*-value | | — | <0.001 | <0.001 | <0.001 | — |

mortality of 50–79% was observed with respect to the concentration of *S. marcescens* cultured on wood. For control samples, termites were mostly alive with low mortality percentage of 16.9%. Based on the results, *S. marcescens* cultured on wood with concentration 10$^6$ CFU ml$^{-1}$ and above are rated to cause 'heavy' mortality effect on *C. curvignathus* under laboratory-controlled condition. At the end of the experiment period, 37% (weight loss) of wood block was consumed in the control sample. For wood blocks treated with *S. marcescens*, the weight loss of 7–22% was recorded.

Generally, termites in all containers with treated wood blocks were active for the first week of the experiment with elaborate tunnelling observed in all glass jars. However, on the third week, termites in treated samples became inactive and no activity was observed on the fourth week of the experiment except treated samples with concentrations 10$^3$ CFU ml$^{-1}$ and below. Most of the termites in containers with treated wood blocks died in the third week of the experiment. Termites in the control sample remained active throughout the whole experiment and established more elaborate tunnel systems (figure 2). Campora & Grace [35] stated that tunnelling in the soil is the main measurement to determine the activeness of termites' colony in a certain area. San & Ling [31] stated that *C. curvignathus* is one of the most aggressive termite species that tend to form extensive tunnel formation. Hapukotuwa & Grace [36] stated that subterranean termites create ramified tunnel systems above or beneath the soil to locate their cellulosic food. Sometimes, these tunnels, which range from tens to hundreds of metres in length,

connect multiple feeding sites. The wood block in the control sample was fully covered by sand in the first week while wood blocks treated with the bacterial concentration higher than $10^9$ CFU ml$^{-1}$ were just partially covered by sand even at the end of the experiment (figure 3). This is another indication of the termite feeding behaviour. Active subterranean termites tend to cover their preferred food source in mud or soil. Subterranean termites like *C. curvignathus* build their nests in soil and are prone to desiccation when they are exposed to the air in the open environment. Therefore, mud tubes or covered runways are constructed when they search for food above ground.

Repellents affect an insect's senses such as smell and taste to prevent it from finding the substrate. The position of termites on the surface indicates the response of the termites to an antagonizing effect such as repellents [32]. At the end of the test, the majority of termites in containers with untreated wood blocks were found beneath the wood surface, whereas in containers with treated wood blocks (bacterial concentrations $10^9$ CFU ml$^{-1}$ and higher), termites' cadavers were found on the block top surface (figure 3). This indicates that a high concentration of *S. marcescens* in treated wood blocks served as a termite-repellent.

As the bacterial-treated wood blocks were meant to be used as toxic baits, it was only considered as an effective bait if it caused high mortality rate from considerable wood consumption by the termites. Toxic bait is meant to attract the termite to feed on it and caused a high termite mortality rate. Although the bacterial concentration of $10^9$ and $10^{12}$ CFU ml$^{-1}$ caused a high mortality rate, the low wood consumption rate (weight loss 7–9%; mean visual rating of 9) could have reflected that the termites were avoiding the wood and the termites may have died of starvation. Chouvenc *et al.* [37] associated this behaviour with the termite's ability to trace the presence of the pathogen in high concentration and try to avoid the area. Wood blocks treated with the bacterial concentration of $10^6$ CFU ml$^{-1}$ are considered to be more effective as a bait compared with other bacterial concentrations used in this study as it caused a high mortality rate (79% termite mortality), and the weight loss and visual rating of the wood showed that the termites had been chewing and biting off a considerable amount of fragments from the wood block (weight loss 15%; mean visual rating of 6.8). Wood treated with the bacterial concentration of $10^6$ CFU ml$^{-1}$ were considered as slow-acting toxic bait, as most termites died in the second week after feeding on the wood block. The cause of death for the termites introduced with wood treated with *S. marcescens* was obviously shown on the cadavers, as the head part of the infected termite turned red, which is the signature colour of *S. marcescens* (figure 3). Termites in the container with wood treated with the bacterial concentration of $10^{12}$ CFU ml$^{-1}$ mostly died in the second week. However, most of the termite cadavers' head did not turn red with very low wood consumption rate which is another indication that the termites died due to starvation instead of infected by *S. marcescens*. Whereas, termites in the container with wood treated with the bacterial concentration of $10^6$ CFU ml$^{-1}$ and below showed the sign of *S. marcescens* infection as most of the heads of the cadavers were red coloured.

*Serratia marcescens* is a Gram-negative bacterium that is unable to produce spores. *Serratia marcescens* have been reported to infect termites and cause septicaemia [10,38,39]. The ability of *S. marcescens* to produce red pigment has become a marker to trace bacterial activity. As shown in this study, *S. marcescens* with its entomopathogenic ability is possibly due to the presence of poly(aniline-co-2,4-diaminophenol) (pADAP) protein in the bacteria [40]. As a response to the release of this bacterial protein, termites expulse the gut contents to the hindgut and release frass pellets [41]. Eventually, this causes the weakening of internal tissues of termite, which may lead to bacterial incursion of the haemocoel, causing septicaemia and death.

Further analysis to determine the efficiency of wood block as the growth medium for *S. marcescens* was carried out. To evaluate the bioavailability of *S. marcescens* on wood over a period of time, wood blocks treated with different bacterial concentrations were placed in different containers containing sterilized sand; similar to termiticidal bioassay tests assembling but without termites. The bacteria-treated wood was removed from the container and the surfaces of the wood were swiped on the nutrient agar to trace the bioavailability of *S. marcescens* on wood every 7 days for the period of 28 days. Results showed that samples treated with the bacterial concentration of $10^3$ CFU ml$^{-1}$ and below were only traceable with *S. marcescens* on the first week of the experiment, but failed to be traced on the third week onwards. While, wood treated with the bacterial concentration of at least $10^6$ CFU ml$^{-1}$ were traceable with *S. marcescens* for at least two weeks.

### 3.1.2. Serratia marcescens cultured in soil

Table 3 shows the results for the termite mortality and its activity towards bacteria-treated soil over the four weeks experiment period. Sand treated with *S. marcescens* showed significant effects on tunnelling

**Table 3.** Behavioural activities and mortality rate of *C. curvignathus* on soil treated with different bacterial concentrations of *S. marcescens*. Values in parentheses are standard deviations. [a–f]Means followed by the same letter in the same column are not significantly different $p \leq 0.05$ according to Tukey's multiple comparison test.

| concentration (CFU ml$^{-1}$) | week | visual rating of tunnelling behaviour | per cent termite mortality [rating] | percentage (%) of wood weight loss [resistance classes] | mean visual rating of wood block | traceability of *S. marcescens* |
|---|---|---|---|---|---|---|
| control | 1st | 3 | 16.9 (1.1)[f] | 37.0 (1.2)[f] | 2.3 (0.8)[d] | no |
|  | 2nd | 4 | [slight] | [very poor] |  | no |
|  | 3rd | 4 |  |  |  | no |
|  | 4th | 4 |  |  |  | no |
| 10$^{12}$ | 1st | 1 | 81.3 (1.4)[a] | 5.4 (1.0)[a] | 9.5 (1.6)[a] | yes |
|  | 2nd | 1 | [heavy] | [resistant] |  | yes |
|  | 3rd | 1 |  |  |  | yes |
|  | 4th | 1 |  |  |  | yes |
| 10$^{9}$ | 1st | 1 | 76.3 (1.4)[b] | 7.4 (1.0)[b] | 8.8 (0.8)[a] | yes |
|  | 2nd | 1 | [heavy] | [resistant] |  | yes |
|  | 3rd | 1 |  |  |  | yes |
|  | 4th | 2 |  |  |  | yes |
| 10$^{6}$ | 1st | 1 | 70.7 (1.3)[c] | 15.7 (0.8)[c] | 6.3 (0.5)[b] | yes |
|  | 2nd | 2 | [heavy] | [poor] |  | yes |
|  | 3rd | 2 |  |  |  | yes |
|  | 4th | 2 |  |  |  | yes |
| 10$^{3}$ | 1st | 2 | 58.2 (1.9)[d] | 25.8 (0.7)[d] | 4.3 (0.4)[c] | yes |
|  | 2nd | 2 | [heavy] | [very poor] |  | yes |
|  | 3rd | 3 |  |  |  | yes |
|  | 4th | 3 |  |  |  | yes |
| 10 | 1st | 2 | 36.2 (1.5)[e] | 30.2 (0.8)[e] | 3.3 (0.3)[c] | yes |
|  | 2nd | 3 | [moderate] | [very poor] |  | yes |
|  | 3rd | 3 |  |  |  | yes |
|  | 4th | 3 |  |  |  | yes |
| *p*-value |  | — | <0.001 | <0.001 | <0.001 | — |

activity, wood consumption and mortality of *C. curvignathus*. The highest mortality rate of 81% was found in samples with termites exposed to soil treated with the bacteria concentration of $10^{12}$ CFU ml$^{-1}$. The lowest termite mortality was found in the glass jar treated with the bacteria concentration of 10 CFU ml$^{-1}$. This suggests that *C. curvignathus* was more tolerant to 10 CFU ml$^{-1}$ bacterial concentration but contact with the bacteria may have some effects on feeding or mortality of termites even with the lowest concentration. *Coptotermes curvignathus* in sand treated with the bacterial concentration of $10^{3}$ and 10 CFU ml$^{-1}$ showed higher mortalities (40% and 62%) and lower wood consumptions than that of control. Control sample with termite mortality of 16.9% was recorded.

Control samples showed elaborate tunnelling in testing jars. Termites did not construct tunnels actively in jars containing sand treated with $10^{12}$ and $10^{9}$ CFU ml$^{-1}$ bacterial concentration. Containers with sand treated with $10^{12}$ and $10^{9}$ CFU ml$^{-1}$ of bacterial concentration might have prevented termites from reaching the wood as tunnelling activity was very minimal with tunnelling activity in less than 50% of the total container area (figure 2). Apparently, the presence of *S. marcescens* in high concentration disrupted termite behaviour and elicited a repellence response away from treated sand.

The wood blocks in containers with bacteria-treated sand were not fully covered with sand, which indicated the feeding route disruption caused by the presence of bacteria in the soil. During the experiment period, some termites in containers with treated sand can be seen crawling on the surface of the treated sand to reach the wood, which is a rarely seen situation. Subterranean termites are rarely found out in the open and do not forage unprotected as they rely on mud tubes and runways which they are tunnelling to protect them from predators.

After about 7 days of exposure, a rapid increase in mortality (greater than 80%) was observed in sand-treated samples with $10^{12}$ CFU ml$^{-1}$ bacterial concentration. Termites remained clumped together at the centre bottom of the container in a cave-like structure with tunnelling activities in less than 25% of the total container area (figure 3). Occasionally, some termites were observed exploring the container with treated sand. Apparently, termites died of starvation due to their inability to cross the treated area to gain access to the wood. Subterranean termites have been shown to live only for about one week without food under laboratory conditions [42].

Signs of nibbling were present on all treated wood blocks and more severe attacks were observed in containers with sand treated with the bacterial concentration of $10^3$ CFU ml$^{-1}$ and below. Comparison of wood consumption in containers with treated sand also indicated lower feeding rate (5.41–30.28%) when compared with control (37%). This suggests an anti-feeding effect of S. marcescens on C. curvignathus. In general, S. marcescens of at least $10^9$ CFU ml$^{-1}$ concentration proved to be toxic for C. curvignathus as termites are inactive even in the first week and tunnelling activities were so minimal (less than 40% of total container area) even at the end of the experiment. Increasing the bacterial concentration resulted in a reduction in tunnelling activity and wood consumption and an increase in mortality. When the bacterium was applied in the soil, termites were highly contacted with the bacteria as soil is the medium for termite activity (a walking platform); this increases the possibility of termites being contacted to the pathogen. Soil treatment with high concentration of S. marcescens may help to create the barrier for exclusion of subterranean termites from building a new tunnel to access the food source, which could be our wooden structures or plantation in the real environment.

Due to the presence of S. marcescens, a large number of the dead termites exhibited a red discoloration and apparently was the cause of the high initial mortality among soil samples introduced with a high concentration of S. marcescens (refer to figure 3). The effect of termites contacted with bacteria through soil medium did not manifest itself immediately after contact but rather slowly within two weeks of exposure. The nature of this contact intoxication effect is not yet clear, but it is possible due to the strong chitinase mechanism and produces high levels of chitinolytic enzymes in S. marcescens which are capable to breakdown the chitin in the insect exoskeleton. A study by Bahar et al. [43] showed a very strong positive connection between the chitinase activities and the insecticidal activities of S. marcescens. The expressed chitinases ChiA, ChiB and ChiC1 from S. marcescens showed toxicity against the honeybee mite, Varroa destructor in the laboratory [9]. Bahar et al. [43] found S. marcescens effective in killing coleopteran insects with more chitin in their exoskeleton. It was reported by Lundgren & Jurat-Fuentes [44] that termite has a weak point which may allow the bacterial entry related to S. marcescens infection at the foregut–midgut junction, although further study would be required to confirm this. The foregut and hindgut of termite are lines with chitinous cuticle, but it is absent in the midgut, which is lined by peritrophic membrane; a matrix of secreted protein fibrils, proteoglycans and glycoproteins function as physical and immunological protection [44]. Serratia marcescens lethality to insect hosts has been shown to have proteolytic and chitinolytic virulence factors that degrade the peritrophic membrane [14]. The zone of transition from the cuticular lining to peritrophic membrane may present a vulnerability to infection which can be exploited by certain bacteria. This could explain the high mortality rate of termites introduced with S. marcescens even just by introducing the bacterium through soil medium.

To evaluate the bioavailability of S. marcescens in the sand throughout the experiment period, sand treated with different bacterial concentrations was placed in different glass containers; similar to termiticidal bioassay tests assembly but without termites. Sand samples were collected from each container every 7 days and were placed on the nutrient agar to trace the bioavailability of S. marcescens in the sand. Results showed that S. marcescens were traceable from all treated sand of different bacterial concentrations even on the fourth week. The S. marcescens-treated soil of at least $10^9$ CFU ml$^{-1}$ concentration was proved to cause heavy effect on the mortality rate with toxicity symptom observed in the first week. Besides, it served as a repellent barrier for C. curvignathus from reaching the wood block with the low weight loss and very light surface damage was observed on the wood block.

**Table 4.** Behavioural activities and mortality rate of *C. curvignathus* on wood block treated with different bacterial concentrations of *P. aeruginosa*. Values in parentheses are standard deviations. [a–f]Means followed by the same letter in the same column are not significantly different $p \leq 0.05$ according to Tukey's multiple comparison test.

| concentration (CFU ml$^{-1}$) | week | visual rating of tunnelling behaviour | per cent termite mortality [rating] | percentage (%) of wood weight loss [resistance classes] | mean visual rating of wood block | traceability of *P. aeruginosa* |
|---|---|---|---|---|---|---|
| control | 1st | 3 | 16.9 (1.1)[f] | 37.0 (1.2)[e] | 2.3 (0.8)[d] | no |
| | 2nd | 4 | [slight] | [very poor] | | no |
| | 3rd | 4 | | | | no |
| | 4th | 4 | | | | no |
| 10$^{12}$ | 1st | 3 | 71.2 (1.6)[a] | 13.4 (0.8)[a] | 8.5 (0.8)[a] | yes |
| | 2nd | 3 | [heavy] | [poor] | | yes |
| | 3rd | 3 | | | | no |
| | 4th | 3 | | | | no |
| 10$^{9}$ | 1st | 3 | 64.3 (1.3)[b] | 15.2 (1.4)[b] | 7.7 (0.8)[a] | yes |
| | 2nd | 3 | [heavy] | [poor] | | yes |
| | 3rd | 3 | | | | no |
| | 4th | 3 | | | | no |
| 10$^{6}$ | 1st | 3 | 57.5 (0.9)[c] | 16.7 (0.8)[c] | 6.2 (0.4)[b] | yes |
| | 2nd | 3 | [moderate] | [poor] | | no |
| | 3rd | 4 | | | | no |
| | 4th | 4 | | | | no |
| 10$^{3}$ | 1st | 3 | 45.5 (1.1)[d] | 19.4 1.1)[c] | 5.7 (0.5)[b] | yes |
| | 2nd | 3 | [moderate] | [very poor] | | no |
| | 3rd | 4 | | | | no |
| | 4th | 4 | | | | no |
| 10 | 1st | 3 | 42.5 (2.2)[e] | 24.7 (1.1)[d] | 4.3 (0.5)[c] | yes |
| | 2nd | 4 | [moderate] | [very poor] | | no |
| | 3rd | 4 | | | | no |
| | 4th | 4 | | | | no |
| *p*-value | | — | <0.001 | <0.001 | <0.001 | — |

## 3.2. Response of *Coptotermes curvignathus* to selective media cultured with *Pseudomonas aeruginosa*

### 3.2.1. *Pseudomonas aeruginosa* cultured on wood

Table 4 shows the results of *C. curvignathus* responded to *P. aeruginosa* cultured on wood as the transfer medium. The highest termite mortality rate of 71% was recorded for samples with wood blocks treated with *P. aeruginosa* at a concentration of 10$^{12}$ CFU ml$^{-1}$. Tunnelling activities were active in all containers with treated wood during the first two weeks of experiment. However, in the third week, termites with wood treated with *P. aeruginosa* at a concentration of 10$^{9}$ CFU ml$^{-1}$ and above become inactive and mostly buried themselves at the bottom of the container, while the termite in control remained active throughout the 28 days of the experiment.

The weight loss of the wood samples treated with different bacterium concentration was ranged from 13% up to 25%. The lowest weight loss of the wood was recorded for the sample treated with the bacteria concentration of 10$^{12}$ CFU ml$^{-1}$. The wood block in the control sample was fully covered by sand in the first week, while wood blocks treated with the bacterial concentration of 10$^{12}$ CFU ml$^{-1}$ were not fully

**Table 5.** Behavioural activities and mortality rate of *C. curvignathus* on soil treated with different bacterial concentrations of *P. aeruginosa*. Values in parentheses are standard deviations. [a–f]Means followed by the same letter in the same column are not significantly different $p \leq 0.05$ according to Tukey's multiple comparison test.

| concentration (CFU ml$^{-1}$) | week | visual rating of tunnelling behaviour | per cent termite mortality [rating] | percentage (%) of wood weight loss [resistance classes] | mean visual rating of wood block | traceability of *P. aeruginosa* |
|---|---|---|---|---|---|---|
| control | 1st | 4 | 16.9 (1.1)[f] | 37.0 (1.2)[c] | 2.3 (0.8)[e] | no |
| | 2nd | 4 | [slight] | [very poor] | | no |
| | 3rd | 4 | | | | no |
| | 4th | 4 | | | | no |
| 10$^{12}$ | 1st | 1 | 63.4 (1.4)[a] | 13.9 (1.5)[a] | 8.8 (1.0)[a] | yes |
| | 2nd | 2 | [moderate] | [poor] | | yes |
| | 3rd | 2 | | | | yes |
| | 4th | 3 | | | | yes |
| 10$^{9}$ | 1st | 1 | 51.6 (1.6)[b] | 12.1 (2.3)[a] | 7.8 (0.5)[a,b] | yes |
| | 2nd | 2 | [moderate] | [poor] | | yes |
| | 3rd | 2 | | | | yes |
| | 4th | 2 | | | | yes |
| 10$^{6}$ | 1st | 2 | 42.5 (1.0)[c] | 14.5 (1.4)[b] | 7.2 (0.4)[b,c] | yes |
| | 2nd | 2 | [moderate] | [poor] | | yes |
| | 3rd | 3 | | | | yes |
| | 4th | 2 | | | | yes |
| 10$^{3}$ | 1st | 2 | 40.8 (1.4)[c] | 19.9 (1.9)[b] | 6.3 (0.4)[c] | yes |
| | 2nd | 3 | [moderate] | [very poor] | | yes |
| | 3rd | 3 | | | | yes |
| | 4th | 3 | | | | yes |
| 10 | 1st | 3 | 35.4 (1.3)[e] | 21.3 (0.8)[b] | 4.7 (0.4)[d] | yes |
| | 2nd | 3 | [moderate] | [very poor] | | yes |
| | 3rd | 3 | | | | no |
| | 4th | 3 | | | | no |
| *p*-value | | — | <0.001 | <0.001 | <0.001 | — |

covered by sand even at the end of the experiment which indicated that the treated woods were less preferable than the untreated wood.

Wood block treated with *P. aeruginosa* at a concentration of $10^{12}$ CFU ml$^{-1}$ can be considered as a functional toxic bait as it attracted the termites to feed on it and caused a high termite mortality rate. It was also considered as slow-acting toxic bait, as most termites died only in the third week after feeding on the wood block. However, the bioavailability of *P. aeruginosa* was not traceable on the second week of the experiment for all treated wood regardless of the concentrations. This could be a disadvantage for *P. aeruginosa*-treated wood to be used as termite bait as the microbes are unable to grow on the wood for a long period of time and unable to retain a useful level of toxicity after one week.

### 3.2.2. *Pseudomonas aeruginosa* cultured on soil

Table 5 shows the data obtained from the experiment using *P. aeruginosa* cultured on soil. Termites exposed to *P. aeruginosa*-treated soil had a lower mortality rate compared with those exposed in samples with treated wood blocks. The highest mortality (63% of termite mortality) using soil as the treatment medium was obtained with the bacteria concentration of $10^{12}$ CFU ml$^{-1}$. Less tunnelling activity on the upper part of containers was observed in samples containing soil with bacterial

suspension. Termites in containers with treated soil mostly remained at the bottom of the container except with the concentration of $10\,\text{CFU}\,\text{ml}^{-1}$ with higher tunnelling activities compared with other concentrations. As explained earlier, this phenomenon may relate to the pathogen-avoiding behaviours by the termite, where termites are trying to avoid contaminated areas by reducing surface contamination. Termites avoid contaminated areas or infected nest-mates and may relocate a nest after an encounter with pathogenic or parasitic microbes [39]. Typically, infected cadavers are isolated and covered with soil, and moribund or dead individuals are quarantined by constructing walls around them with faecal pellets or soil [45]. These behaviours help to diminish the probability of infection spreading and may be triggered by rapid changes in the cuticular chemistry of infected or dead nest-mates [46].

Termites in the containers with soil treated with *P. aeruginosa* feed very little on the wood when compared with the control sample. However, higher weight loss and much severe visual damage on wood blocks were observed in the samples with treated soil when compared with samples with *P. aeruginosa*-treated wood. Tunnelling activity by the termites was active for the beginning of the experiment and became inactive in the third week and almost no tunnelling activity was observed in the fourth week of the experiment. This shows that the detrimental effects of *P. aeruginosa* towards termites were only traceable on the third week after the termites were introduced to the treated soil. Sand samples were collected from each container every 7 days and were placed on the nutrient agar to trace the bioavailability of *P. aeruginosa* in the sand. Results showed that *P. aeruginosa* was traceable from all treated sand of different bacterial concentrations even on the fourth week except samples with the bacterial concentration of $10\,\text{CFU}\,\text{ml}^{-1}$ which was not traceable on the third week. The *P. aeruginosa*-treated soil of at least $10^6\,\text{CFU}\,\text{ml}^{-1}$ concentration was proved to cause a moderate effect on the mortality rate with slow toxicity symptom and unable to act as a complete repellent barrier for *C. curvignathus* from reaching the wood block.

*Pseudomonas aeruginosa* is a Gram-negative bacterium that can inhibit the growth of a wide range of organisms and is pathogenic to mammals [47], insects [48] and nematodes [49]. Hydrogen cyanide is a secondary metabolite of *P. aeruginosa* [50], which is believed to be responsible for the entomopathogenic potential of this bacteria [51,52]. Devi & Kothamasi [53] demonstrated that cyanide of bacterial origin may constrain cytochrome *c* oxidase (CCO) of the termite *Odontotermes obesus* respiratory chain and proved that *Pseudomonas* spp., a hydrogen cyanide producing bacteria can actually kill a macroscopic insect pest by cyanide poisoning. This ability of pseudomonad metabolites such as cyanide represents an attractive possibility for insect pest management as it may overcome the behavioural adaptations of social insect pests such as termites, by blocking the termite respiration instead of through infection or predation. This may explain the considerably high mortality rate of termites introduced with *P. aeruginosa* via bacterial-treated wood blocks, which may due to the combination of ingestion and contact intoxication.

## 4. Conclusion

This study was conducted to understand the mortality rate and behaviour changes of *C. curvignathus* introduced with *S. marcescens* and *P. aeruginosa* of different concentrations using wood and soil as the transfer media. In general, a higher concentration of bacteria in soil caused a reduction in tunnelling activity and wood consumption and an increase in mortality. This could be problematic considering that potential application would require a large volume of bacteria to control tunnelling and feeding of *C. curvignathus*. However, application on wood revealed a different outcome. Instead of applying higher bacterial concentration to create a better effect, wood treated with *S. marcescens* of $10^6\,\text{CFU}\,\text{ml}^{-1}$ concentration proved to be more efficient as a bait compared with applications with higher concentration as it creates a higher mortality rate and it is still able to attract the termites to feed on the wood. Wood or soil treated with *S. marcescens* of $10^9\,\text{CFU}\,\text{ml}^{-1}$ concentration and above generally creates a high toxicity or repellent barrier for *C. curvignathus* from feeding on the wood. Wood treated with *P. aeruginosa* of at least $10^9\,\text{CFU}\,\text{ml}^{-1}$ concentrations acted as a slow-acting toxic bait to cause a high termite mortality rate due to toxic feeding and does not serve as a good repellent to prevent termites from feeding on the wood. However, the ability for *S. marcescens* and *P. aeruginosa* to survive on wood is low, which means the bait is unable to retain a useful level of toxicity for a long period of time and frequent reapplication is needed. The lower mortality rate from *P. aeruginosa*-treated soil compared with treated wood indicated that the toxicity effect was due to the ingestion factor rather than the physical contact factor. *Coptotermes curvignathus* behavioural activities on *S. marcescens* and *P. aeruginosa* were better understood from this research, thus, other cellulosic bait matrix should be

tested for their functionality as bacterial transfer medium; prolonging survival of bacteria and also highly palatable for *C. curvignathus*. To verify these laboratory results applicable to a real situation, further research on the application strategies of *S. marcescens* and *P. aeruginosa* as effective biotermiticides has to be conducted in the natural environment of *C. curvignathus* habitat. Authentic information on the effects of the biotermiticides and understanding between the application methods and the potential hazard to human or animal systems need further investigations before recommending a termite management method. These additional studies are required to develop this approach as a feasible, sustainable and environmentally sound biological termite management method.

Ethics. We declare that the work submitted for the publication is original, has not been published elsewhere, accepted for publication elsewhere or under editorial review for publication elsewhere; and that all the authors mutually agree with its content and have approved the paper for release and submission. The manuscript does not contain experiments using animals. At the same time, the manuscript does not contain human studies.
Data accessibility. Macroscopic morphological observation of bacteria isolates, termite mortality percentage, presence of introduced bacteria on infected termite and Biocontrol score data file have been uploaded to the Dryad Digital Repository: https://doi.org/10.5061/dryad.66t1g1k07 [54].
Authors' contributions. K.L.C. carried out data collection, participated in analysis and interpretation of data and drafted the article; P.S.H. agreed to be accountable for all aspects of the work in ensuring that questions related to the accuracy or integrity of any part of the work are appropriately investigated and resolved; A.C.L. and Z.A edited and revised the article critically for important intellectual content; C.L.L. participated in analysis and interpretation of data; P.S.K. participated in analysis and interpretation of data; W.Z.W. and W.Z.G. carried out data collection. All authors gave final approval for publication.
Competing interests. We declare we have no competing interests.
Funding. The authors are grateful for the financial support from the co-author P.S.H. under the Higher Institution Centre of Excellence (HICoE) (grant no. 6369110) project at the Institute of Tropical Forestry and Forest Products which was given by the Ministry of Higher Education Malaysia (MOHE).
Acknowledgements. The authors thank all project members for support and collaboration. The authors also sincerely thank the postgraduate students who participated in the field sampling exercise.

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
