## [Peer Review File · Royal Society Open Science]

Review History

RSOS-201311.R0 (Original submission)

Review form: Reviewer 1

Is the manuscript scientifically sound in its present form?

Yes

Are the interpretations and conclusions justified by the results?

No

Is the language acceptable?

Yes

Do you have any ethical concerns with this paper?

No

Have you any concerns about statistical analyses in this paper?

Yes

Recommendation?

Major revision is needed (please make suggestions in comments)

Comments to the Author(s)

RSOS-201311

Application strategies by selective medium treated with entomopathogenic bacteria *Serratia marcescens* and *Pseudomonas aeruginosa* as potential biocontrol against *Coptotermes curvignathus*

Different concentrations of *Serratia marcescens* and *Pseudomonas aeruginosa* were incorporated and treated onto the soil and wood respectively and found out their impact on the wood consumption and mortality of *Coptotermes curvignathus*.

My main concerns are listed here:

1. Main focus of the work is management of *Coptotermes curvignathus*. But nothing has been included in the introduction, I suggest to include economic importance, management practices so far followed, and what is necessary of the proposed work would be included.
2. On what basis authors used *Serratia marcescens* and *Pseudomonas aeruginosa* against *Coptotermes curvignathus* should be included in the introduction section
3. The ability for *S. marcescens* and *P. aeruginosa* to survive on wood could be specified in the ms.
4. Results is included in the introduction (line 47, 48) should be removed rather, precise and appropriate objectives can be included
5. In page 3, it was clearly mentioned that isolated from soil on which *C. curvignathus* forage from, at the same time in page 4, Bacterial strains from stock cultures were transferred. Whether authors used the former or latter isolate for the proposed study?
6. Line 11, confirm its concentration, elaborate
7. Line 15, specify the name of the termite infested trees
8. Line 33, washed and dried sand?
9. On what basis authors considered 1: 5 ratio of soldiers : workers
10. Experimental conditions like humidity, light should be included
11. Termite mortality was recorded only after the experiments (28 days), then what you observed on 7, 14, and 21 days of observations
12. Tunnelling activity and behavioural activities of termites, specify what are the behaviors observed
13. Throughout the ms, authors included table data (Eg: 50 - 79%, 40% and 62%). I suggest analysing the data with multivariate statistical analyses, and include df, F and P values for the better scientific expression of results.
14. Why not tunneling has been altered or arrested would be discussed properly
15. From figure 2, it is very clear that authors have not maintained a constant amount of soil or wood/height of soil or wood for the various treatments? specific reasons if any should be specified
16. antagonizing effect such as repellents, how repellent activity is considered to be an antagonizing effect?
17. Table 1 deals with rating, however, it has not been included in the result section rather in tables 2-4. If I am not correct, kindly check?
18. Table 2, 3,4, 5 mean with SD or SE should be included

Other suggestions to be carried out before acceptance of the manuscript

1. Page 3, line 33, since first time authors using *C. curvignathus*, genus name, author with taxonomical details should be provided
2. Page 3, line 35-37, irrelevant sentence, remove it.

3. Page 3, line 59-60, is the microbes obtained from normal or infected *C. curvignathus*?
4. (latitude 3.16°N and 101.56°E longitude) to (3.16°N and 101.56°E).
5. Reference no. 5, Line 45, *curvignathus's* to *curvignathus*
6. Reference no. 22, *Curvignathus* to *curvignathus*

Review form: Reviewer 2

Is the manuscript scientifically sound in its present form?

Yes

Are the interpretations and conclusions justified by the results?

Yes

Is the language acceptable?

Yes

Do you have any ethical concerns with this paper?

No

Have you any concerns about statistical analyses in this paper?

Yes

Recommendation?

Accept with minor revision (please list in comments)

Comments to the Author(s)

The manuscript presents promising preliminary results of using two bacteria species as a biotermicides against *Coptotermes curvignathus*. I have some minor comments to the text.

It would be good to add information to the introduction what metabolites and enzymes, with termicide potential, produce *Pseudomonas* and *Serratia* spp. For example, *P.aeruginosa* is most known from producing pyocyanin.

The text needs English grammar and style checking as there are linguistic errors, e.g. Line 50/51 page 3 should be „media“ instead of „mediums“,

Please check and explain all acronyms used in the text, e.g. pADAP.

How many samples for each bacteria dose were investigated?

Review form: Reviewer 3

Is the manuscript scientifically sound in its present form?

No

Are the interpretations and conclusions justified by the results?

No

Is the language acceptable?

No

Do you have any ethical concerns with this paper?

No

Have you any concerns about statistical analyses in this paper?

Yes

Recommendation?

Major revision is needed (please make suggestions in comments)

Comments to the Author(s)

The manuscript "Application strategies by selective médium treated with entomopathogenic bacteria as potential biocontrol against *Coptotermes curvignathus*" by Chin et al. describes bioassays results from lab conditions concerning effects of *S. marcescens* and *P. aeruginosa* on survival of *C. curvignathus*. Biocontrol agents are environmental friendly and need to be explored for future uses, as alternative to chemicals in pest controlling. Though the methods seem to be well conducted for this kind of evaluation, the data are not well analysed. I missed statistical analysis of the data, especially concerning the weight of the wooden blocks which are not data based on notes (which might be subjective). But also the notes with percentages should be tested statistically before the conclusions. Also, means without standard deviations are not informative. The way results are present is too descriptive and lacks support for the findings. I suggest rejection in its present form, but with some statistics the analysis should improve the evidences and give more support for the findings, which are quite interesting. So my main suggestion is to properly analyze the data with statistical tests in order to have support for the conclusions.

Minor comments?

Line 1 page 5 ...experiments were kept in total darkness for 30 days. But the evaluations were ended at 28th day.

There is misleading of British and American English styles throughout the text that should be revised by the authors and/or a native speaker (I am not a native english speaker myself and I understand the difficulties, and if I was able to detect these mistakes, then the text should be revised). For example, in some sentences the word behavior is written in American stlyle (line 33/34 page 5) and in others in British style - behaviour (line 17 page 6).

Decision letter (RSOS-201311.R0)

Dear Dr LEE

The Editors assigned to your paper RSOS-201311 "Application strategies by selective medium treated with entomopathogenic bacteria as potential biocontrol against *Coptotermes curvoignathus*" have now received comments from reviewers and would like you to revise the paper in accordance with the reviewer comments and any comments from the Editors. Please note this decision does not guarantee eventual acceptance.

Please submit your revised manuscript and required files (see below) no later than 21 days from today's (ie 16-Nov-2020) date. Note: the ScholarOne system will 'lock' if submission of the revision is attempted 21 or more days after the deadline. If you do not think you will be able to meet this deadline please contact the editorial office immediately.

on behalf of Dr Berat Haznedaroglu (Associate Editor) and Pete Smith (Subject Editor)
openscience@royalsociety.org

Reviewer comments to Author:

Reviewer: 1

Comments to the Author(s)

RSOS-201311

Application strategies by selective medium treated with entomopathogenic bacteria *Serratia marcescens* and *Pseudomonas aeruginosa* as potential biocontrol against *Coptotermes curvignathus*

Different concentrations of *Serratia marcescens* and *Pseudomonas aeruginosa* were incorporated and treated onto the soil and wood respectively and found out their impact on the wood consumption and mortality of *Coptotermes curvignathus*.

My main concerns are listed here:

1. Main focus of the work is management of *Coptotermes curvignathus*. But nothing has been included in the introduction, I suggest to include economic importance, management practices so far followed, and what is necessary of the proposed work would be included.
2. On what basis authors used *Serratia marcescens* and *Pseudomonas aeruginosa* against *Coptotermes curvignathus* should be included in the introduction section
3. The ability for *S. marcescens* and *P. aeruginosa* to survive on wood could be specified in the ms.

4. Results is included in the introduction (line 47, 48) should be removed rather, precise and appropriate objectives can be included
 5. In page 3, it was clearly mentioned that isolated from soil on which *C. curvignathus* forage from, at the same time in page 4, Bacterial strains from stock cultures were transferred. Whether authors used the former or latter isolate for the proposed study?
 6. Line 11, confirm its concentration, elaborate
 7. Line 15, specify the name of the termite infested trees
 8. Line 33, washed and dried sand?
 9. On what basis authors considered 1: 5 ratio of soldiers : workers
 10. Experimental conditions like humidity, light should be included
 11. Termite mortality was recorded only after the experiments (28 days), then what you observed on 7, 14, and 21 days of observations
 12. Tunnelling activity and behavioural activities of termites, specify what are the behaviors observed
 13. Throughout the ms, authors included table data (Eg: 50 – 79%, 40% and 62%). I suggest analysing the data with multivariate statistical analyses, and include df, F and P values for the better scientific expression of results.
 14. Why not tunneling has been altered or arrested would be discussed properly
 15. From figure 2, it is very clear that authors have not maintained a constant amount of soil or wood/height of soil or wood for the various treatments? specific reasons if any should be specified
 16. antagonizing effect such as repellents, how repellent activity is considered to be an antagonizing effect?
 17. Table 1 deals with rating, however, it has not been included in the result section rather in tables 2-4. If I am not correct, kindly check?
 18. Table 2, 3,4, 5 mean with SD or SE should be included
- Other suggestions to be carried out before acceptance of the manuscript
1. Page 3, line 33, since first time authors using *C. curvignathus*, genus name, author with taxonomical details should be provided
 2. Page 3, line 35-37, irrelevant sentence, remove it.
 3. Page 3, line 59-60, is the microbes obtained from normal or infected *C. curvignathus*?
 4. (latitude 3.16°N and 101.56°E longitude) to (3.16°N and 101.56°E).
 5. Reference no. 5, Line 45, *curvignathus*'s to *curvignathus*
 6. Reference no. 22, *Curvignathus* to *curvignathus*

Reviewer: 2

Comments to the Author(s)

The manuscript presents promising preliminary results of using two bacteria species as a biotermiticides against *Coptotermes curvignathus*. I have some minor comments to the text.

It would be good to add information to the introduction what metabolites and enzymes, with termiticide potential, produce *Pseudomonas* and *Serratia* spp. For example, *P. aeruginosa* is most known from producing pyocyanin.

The text needs English grammar and style checking as there are linguistic errors, e.g. Line 50/51 page 3 should be „media” instead of „mediums”,

Please check and explain all acronyms used in the text, e.g. pADAP.

How many samples for each bacteria dose were investigated?

Reviewer: 3

Comments to the Author(s)

The manuscript “Application strategies by selective médium treated with entomopathogenic bacteria as potential biocontrol against *Coptotermes curvignathus*” by Chin et al. describes

bioassays results from lab conditions concerning effects of *S. marcescens* and *P. aeruginosa* on survival of *C. curvignathus*. Biocontrol agents are environmental friendly and need to be explored for future uses, as alternative to chemicals in pest controlling. Though the methods seem to be well conducted for this kind of evaluation, the data are not well analysed. I missed statistical analysis of the data, especially concerning the weight of the wooden blocks which are not data based on notes (which might be subjective). But also the notes with percentages should be tested statistically before the conclusions. Also, means without standard deviations are not informative. The way results are present is too descriptive and lacks support for the findings. I suggest rejection in its present form, but with some statistics the analysis should improve the evidences and give more support for the findings, which are quite interesting. So my main suggestion is to properly analyze the data with statistical tests in order to have support for the conclusions.

Minor comments?

Line 1 page 5 ...experiments were kept in total darkness for 30 days. But the evaluations were ended at 28th day.

There is misleading of British and American English styles throughout the text that should be revised by the authors and/or a native speaker (I am not a native english speaker myself and I understand the difficulties, and if I was able to detect these mistakes, then the text should be revised). For example, in some sentences the word behavior is written in American style (line 33/34 page 5) and in others in British style - behaviour (line 17 page 6).

===PREPARING YOUR MANUSCRIPT===

===PREPARING YOUR REVISION IN SCHOLARONE===

Author's Response to Decision Letter for (RSOS-201311.R0)

See Appendix A.

RSOS-201311.R1 (Revision)

Review form: Reviewer 2

Is the manuscript scientifically sound in its present form?

Yes

Are the interpretations and conclusions justified by the results?

Yes

Is the language acceptable?

Yes

Do you have any ethical concerns with this paper?

No

Have you any concerns about statistical analyses in this paper?

No

Recommendation?

Accept as is

Comments to the Author(s)

All comments were addressed and text improved according to them.

Review form: Reviewer 3

Is the manuscript scientifically sound in its present form?

No

Are the interpretations and conclusions justified by the results?

Yes

Is the language acceptable?

No

Do you have any ethical concerns with this paper?

No

Have you any concerns about statistical analyses in this paper?

Yes

Recommendation?

Accept with minor revision (please list in comments)

Comments to the Author(s)

Abstract - Line 35-36 - *Coptotermes curvignathus* Holmgren should be out of parentheses

Page 7, Line 51 – Termite damage is a major problem in the tropics that causes...

I think that it still contain English grammar mistakes. The authors should revise it prior to be considered for publication. Another examples: Page 8, Line 15/16 – The conventional termiticide method... or The conventional termiticide methods...; Page 8, Lines 27/28 – termiticide bait method or termiticide bait methods.; and Page 9, lines 4/5 – Both media, because mediums mean plural of spiritual people. I've already corrected that in the first review.

Page 11 – Lines 35-40 – Lines 55/56 Considering statistics, I would use a generalized linear model to increase statistical power, analyzing altogether the termite mortality, wood block visual rating and weight loss, instead of different ANOVA + Tukey (as results shown in table 2).

Decision letter (RSOS-201311.R1)

Dear Dr LEE

On behalf of the Editors, we are pleased to inform you that your Manuscript RSOS-201311.R1 "Application strategies by selective medium treated with entomopathogenic bacteria as potential biocontrol against *Coptotermes curvignathus*" has been accepted for publication in Royal Society Open Science subject to minor revision in accordance with the referees' reports. Please find the referees' comments along with any feedback from the Editors below my signature.

Please submit your revised manuscript and required files (see below) no later than 7 days from today's (ie 04-Mar-2021) date. Note: the ScholarOne system will 'lock' if submission of the revision is attempted 7 or more days after the deadline. If you do not think you will be able to meet this deadline please contact the editorial office immediately.

Please note article processing charges apply to papers accepted for publication in Royal Society Open Science (<https://royalsocietypublishing.org/rsos/charges>). Charges will also apply to papers transferred to the journal from other Royal Society Publishing journals, as well as papers submitted as part of our collaboration with the Royal Society of Chemistry

(<https://royalsocietypublishing.org/rsos/chemistry>). Fee waivers are available but must be requested when you submit your revision (<https://royalsocietypublishing.org/rsos/waivers>).

on behalf of Dr Berat Haznedaroglu (Associate Editor) and Pete Smith (Subject Editor)
openscience@royalsociety.org

Reviewer comments to Author:

Reviewer: 2

Comments to the Author(s)

All comments were addressed and text improved according to them.

Reviewer: 3

Comments to the Author(s)

Abstract - Line 35-36 - *Coptotermes curvignathus* Holmgren should be out of parentheses

Page 7, Line 51 – Termite damage is a major problem in the tropics that causes...

I think that it still contain English grammar mistakes. The authors should revise it prior to be considered for publication. Another examples: Page 8, Line 15/16 – The conventional termiticide method... or The conventional termiticide methods...; Page 8, Lines 27/28 – termiticide bait method or termiticide bait methods.; and Page 9, lines 4/5 – Both media, because mediums mean plural of spiritual people. I've already corrected that in the first review.

Page 11 – Lines 35-40 – Lines 55/56 Considering statistics, I would use a generalized linear model to increase statistical power, analyzing altogether the termite mortality, wood block visual rating and weight loss, instead of different ANOVA + Tukey (as results shown in table 2).

===PREPARING YOUR MANUSCRIPT===

===PREPARING YOUR REVISION IN SCHOLARONE===

<https://royalsociety.org/journals/authors/author-guidelines/#data>. You should ensure that

you cite the dataset in your reference list. If you have deposited data etc in the Dryad repository, please only include the 'For publication' link at this stage. You should remove the 'For review' link.

Author's Response to Decision Letter for (RSOS-201311.R1)

See Appendix B.

Decision letter (RSOS-201311.R2)

Dear Dr LEE,

It is a pleasure to accept your manuscript entitled "Application strategies by selective medium treated with entomopathogenic bacteria as potential biocontrol against *Coptotermes curvignathus*" in its current form for publication in Royal Society Open Science.

Please ensure that you send to the editorial office your 'for publication/acceptance' version of your Dryad data deposition, so this may be included in your proof (the Dryad link you have supplied is the 'for review/private' version, and we will require the public version URL).

Please see the Royal Society Publishing guidance on how you may share your accepted author manuscript at <https://royalsociety.org/journals/ethics-policies/media-embargo/>. After

publication, some additional ways to effectively promote your article can also be found here <https://royalsociety.org/blog/2020/07/promoting-your-latest-paper-and-tracking-your-results/>.

on behalf of Dr Berat Haznedaroglu (Associate Editor) and Pete Smith (Subject Editor)
openscience@royalsociety.org

Appendix A

Response to Reviewer's Comments Manuscript RSOS-201311

We thank the reviewers for their thoughtful reviews and valuable comments of our manuscript. We take concerns seriously and have addressed them to the best of our abilities. Changes have been made as suggested by the reviewers and were highlighted in green in the manuscript. Some of the more notable changes are listed as below;

Reviewer:

1

Reviewer 1:

Comments to the Author(s):

1. Main focus of the work is management of *Coptotermes curvignathus*. But nothing has been included in the introduction, I suggest to include economic importance, management practices so far followed, and what is necessary of the proposed work would be included.

Response: Changes have been made in the Introduction as suggested by the reviewer.

2. On what basis authors used *Serratia marcescens* and *Pseudomonas aeruginosa* against *Coptotermes curvignathus* should be included in the introduction section.

Response: Changes have been made in the Introduction as suggested by the reviewer.

3. Results is included in the introduction (line 47, 48) should be removed rather, precise and appropriate objectives can be included.

Response: Changes have been made in the Introduction as suggested by the reviewer.

4. In page 3, it was clearly mentioned that isolated from soil on which *C. curvignathus* forage from, at the same time in page 4, Bacterial strains from stock cultures were transferred. Whether authors used the former or latter isolate for the proposed study?

Response: Changes have been made and highlighted in the manuscript to clarify the method used.

Isolated bacterial strains were maintained as stock cultures and were stored on a cool, low light shelf. These stock cultures were maintained by re-plating each onto new nutrient media at least once a month depending on the nature of each isolated strain.

5. Line 11, confirm its concentration, elaborate.

Response: Changes have been made and highlighted in the manuscript to clarify the method used.

At the same time, the purity and colony forming units for the cultures were checked to confirm its concentration by using pour-plating and spread-plating methods.

6. Line 15, specify the name of the termite infested tree.

Response: Changes have been made as suggested by the reviewer and highlighted in the manuscript.

Subterranean termites, C. curvignathus were collected from termite infested rubber tree (Hevea brasiliensis)

7. Line 33, washed and dried sand?

Response: Sand used in the termite test has to be cleaned, washed and sterilized to remove organic matters and potentially harmful bacteria that might affect the results of the test. Changes have been made and highlighted in the manuscript to clarify the method used.

8. On what basis authors considered 1: 5 ratio of soldiers : workers

Response: This ratio is according to the American Wood Preservers' Association standard (AWPA) E1-09. Soldiers are a much more specialized caste, and make up only about 10% of any termite colony.

9. Experimental conditions like humidity, light should be include.

Response: Changes have been made as suggested by the reviewer and highlighted in the manuscript.

250 workers and 25 soldiers of C. curvignathus were later released in each of the assembled glass container and kept in an incubator at 28°C in total darkness for 28 days. These glass containers were kept in a moist condition (70 – 80% relative humidity) for four weeks.

10. Termite mortality was recorded only after the experiments (28 days), then what you observed on 7, 14, and 21 days of observations.

Response: Changes have been made and highlighted in the manuscript to clarify the method used.

These containers were examined every 7 days for 28 days, to record for the presence of tunnelling and position of the termites in the container. At the end of the experiment, all wood blocks were removed and oven dried at 90°C for three days and allowed to cool to

room temperature in a desiccator for one hour before weighing. The mortality of the termites was also recorded at the end of the experiment.

11. Tunnelling activity and behavioural activities of termites, specify what are the behaviors observed.

Response: Changes have been made and highlighted in the manuscript to clarify the method used. The behaviours observations methods were stated in Table 1: *Rating system for evaluations of termite mortality, behavioural observations, and wood block mass loss.*

12. Throughout the ms, authors included table data (Eg: 50 – 79%, 40% and 62%). I suggest analysing the data for the better scientific expression of results.

Response: Changes have been made as suggested by the reviewer. All the data was analysed and tabulated in Table 2 – Table 5.

All statistical analyses were performed using the SAS software (Version 9.0). The data on termite mortality, wood block visual rating and weight loss for each treatment were analysed using ANOVA at 95% confident level ($p < 0.05$). Tukey–Kramer’s multiple comparison test was applied to analyse the differences of the bacterial suspension concentration effect when significance was observed. The effects were considered not statistically significant when the p -value was higher than 0.05 at 95% confidence level.

13. From figure 2, it is very clear that authors have not maintained a constant amount of soil or wood/height of soil or wood for the various treatments? specific reasons if any should be specified

Response: Constant amount (150 gram) of sand was added in each jar. Some jar with higher tunnelling activities termites causing the sand to be loosely compacted, while, jars with lower tunnelling activities, sand was more compact. Tunnelling usually includes excavation, loading, transportation and deposition of soil particles (Lee and Wood, 1971). When the condition is permitting in the testing jar, termites penetrated the sand and immediately started their foraging activity from the bottom centre of the jar, excavating and transporting sand particles and building galleries in the jars. The excavation and transport of sand left hollow spaces at the bottom of the jar and hollowing continued upwards as the excavation continued. The sand either remained intact above the hollow created at the bottom of the jar or collapsed under its own weight or the weight of wood block sitting on top. A thin layer of soil was formed at the top of the wood block in testing jars with active termite activities. Termites also spread the sand particles all over the surface of the jar to help them move up the jar wall. Some of the explanations have been included in the manuscript to clarify the changes.

14. Antagonizing effect such as repellents, how repellent activity is considered to be an antagonizing effect?

Response: Repellents affect an insect's senses such as smell and taste to prevent it from finding the substrate.

15. Table 1 deals with rating, however, it has not been included in the result section rather in tables 2-4. If I am not correct, kindly check?

Response: Table 1 explains the rating system used in this study so we placed Table 1 in the methodology section instead in the result section. The results obtained by using the rating described in Table 1 were tabulated in Table 2 – Table 5 and further discussed in the Results and Discussion section.

Termite mortality, behavioural observations, and wood block mass loss were recorded using the rating system as shown in Table 1: Rating system for evaluations of termite mortality, behavioural observations, and wood block mass loss.

16. Table 2, 3, 4, 5 mean with SD or SE should be included

Response: Changes have been made as suggested by the reviewer. Standard deviations were added and tabulated in Table 2 – Table 5.

Reviewer: 2

Comments to the author(s):

1. It would be good to add information to the introduction what metabolites and enzymes, with termicide potential, produce *Pseudomonas* and *Serratia* spp.

Response: Changes have been made in the Introduction as suggested by the reviewer.

2. The text needs English grammar and style checking as there are linguistic errors, e.g. Line 50/51 page 3 should be “media” instead of “mediums”

Response: Corrections have been made and were highlighted in the manuscript. The paper has also been sent to a native English speaker and has been carefully revised to improve grammar and readability.

3. Please check and explain all acronyms used in the text, e.g. pADAP.

Response: Changes have been made as suggested by the reviewer.

4. How many samples for each bacteria dose were investigated?

Response: Changes have been made and highlighted in the manuscript to clarify the method used.

All treatments were replicated 6 times.

Reviewer: 3

Comments to the Author(s)

1. Notes with percentages should be tested statistically before the conclusions. Also, means without standard deviations are not informative. Properly analyze the data with statistical tests in order to have support for the conclusions.

Response: Changes have been made as suggested by the reviewer. All the data was analysed and tabulated in Table 2 – Table 5. Standard deviations were also added.

All statistical analyses were performed using the SAS software (Version 9.0). The data on termite mortality, wood block visual rating and weight loss for each treatment were analysed using ANOVA at 95% confident level ($p < 0.05$). Tukey–Kramer’s multiple comparison test was applied to analyse the differences of the bacterial suspension concentration effect when significance was observed. The effects were considered not statistically significant when the p -value was higher than 0.05 at 95% confidence level.

2. There is misleading of British and American English styles throughout the text that should be revised by the authors and/or a native speaker.

Response: Corrections have been made and were highlighted in the manuscript. The paper has also been sent to a native English speaker and has been carefully revised to improve grammar and readability.

Appendix B

Response to Reviewer's Comments Manuscript RSOS-201311

We thank the reviewers for their thoughtful reviews and valuable comments of our manuscript. We take concerns seriously and have addressed them to the best of our abilities. Changes have been made as suggested by the reviewers and were highlighted in green in the manuscript. Some of the more notable changes are listed as below;

Reviewer 2:

1. All comments were addressed and text improved.

Response: Thank you very much for reviewing our manuscript.

Reviewer 3:

Comments to the Author(s):

1. Abstract - Line 35-36 - *Coptotermes curvignathus* Holmgren should be out of parentheses.

Response: Changes have been made and highlighted in the manuscript.

2. Page 7, Line 51 – Termite damage is a major problem in the tropics that causes...I think that it still contains English grammar mistakes. The authors should revise it prior to be considered for publication. Another examples: Page 8, Line 15/16 – The conventional termiticide method... or The conventional termiticide methods...; Page 8, Lines 27/28 – termiticide bait method or termiticide bait methods...; and Page 9, lines 4/5 – Both media, because mediums mean plural of spiritual people. I've already corrected that in the first review.

Response: Changes have been made and highlighted in the manuscript.

3. Page 11 – Lines 35-40 – Lines 55/56 Considering statistics, I would use a generalized linear model to increase statistical power, analyzing altogether the termite mortality, wood block visual rating and weight loss, instead of different ANOVA + Tukey (as results shown in table 2).

Response: We thank the reviewer for these suggestions, however, to analyse altogether the dependent variables (termite mortality, wood block visual rating and weight loss), the dependent variables have to be correlated which is not in this case. Hence, separate ANOVAs are more appropriate to be applied in this study. We believe the statistical analysis used is sufficient to draw meaningful interpretation and in simplifying data so that their significance is comprehensible.